# Iron-catalyzed fluoroalkylative alkylsulfonylation of alkenes via radical-anion relay

Xiaoya Hou[1], Hongchi Liu[1] & Hanmin Huang ® [1,2] ✉

Transition metal-catalyzed reductive difunctionalization of alkenes with alkyl halides is a powerful method for upgrading commodity chemicals into densely functionalized molecules. However, super stoichiometric amounts of metal reductant and the requirement of installing a directing group into alkenes to suppress the inherent β-H elimination bring great limitations to this type of reaction. We demonstrate herein that the difunctionalization of alkenes with two different alkyl halides is accessible via a radical-anion relay with $Na_2S_2O_4$ as both reductant and sulfone-source. The $Na_2S_2O_4$ together with the electron-shuttle catalyst is crucial to divert the mechanistic pathway toward the formation of alkyl sulfone anion instead of the previously reported alkylmetal intermediates. Mechanistic studies allow the identification of carbon-centered alkyl radical and sulfur-centered alkyl sulfone radical, which are in equilibrium via capture or extrusion of $SO_2$ and could be converted to alkyl sulfone anion accelerated by iron electron-shuttle catalysis, leading to the observed high chemoselectivity.

Olefins serve as a sought-after class of feedstocks for chemical synthesis by virtue of their wide existence, availability as well as versatility[1–3]. The intermolecular difunctionalization of olefins by introducing two fragments across the C=C double bond represents a powerful tool for upgrading the feedstocks into structurally complex and densely functionalized molecules[4–9]. As an important subset of these transformations, the metal-catalyzed reductive dicarbo-functionalization of olefins with two different carbon electrophiles avoids the preparation of hazardous organometallics, thus has emerged as an efficient method for the simultaneous construction of two carbon–carbon bonds in one single operation[10–15]. However, due to the presence of alkylmetal species as well as the involvement of classical oxidative addition and reductive elimination in the catalytic cycle, the metal-catalyzed reductive olefin-difunctionalization reaction still mechanistically suffers from undesired β-H elimination and is reluctant to reductive elimination (Fig. 1a, path a)[16–18]. The remaining challenges include: (1) excess amounts of metal (Zn, Mn, Mg, etc.) were required as reductant to maintain the catalytic cycle, endowing the reaction atom-uneconomical and environment-unfriendly[10–19]; (2) directing groups were typically required for

olefins to inhibit the competing β-H elimination of alkylmetal species with attenuated reductive elimination activity, significantly limiting the substrate scope and reducing the versatility and step-economy of the protocols[20–23]. Therefore, developing a new strategy for the reductive olefin-difunctionalization that addresses these challenges is in urgent demand and described. Our approach to this goal was based on the fact that the carbon-centered radical **A** was generally generated in most of the metal-catalyzed reductive alkene-difunctionalization reactions (Fig. 1a, path a). We hypothesized that once the newly formed carbon-centered radical **A** was captured by a suitable radical anion or its equivalent rather than by the metal-catalyst, the nucleophilic anion **B** would be generated and react with another carbon electrophile via simple nucleophilic substitution (Fig. 1a, path b), furnishing the difunctionalization of alkenes with two different alkyl electrophiles. As such, identifying a suitable radical anion or its equivalent is a prerequisite for the realization of the above reaction.

Sodium dithionite ($Na_2S_2O_4$) is not only commonly employed as a single-electron reducing agent in metalloenzyme chemistry but also can be utilized as a precursor of sulfoxylate anion radical[24–28]. Previous

[1]Key Laboratory of Precision and Intelligent Chemistry and Department of Chemistry, University of Science and Technology of China, Hefei, P. R. China. [2]Key Laboratory of Green and Precise Synthetic Chemistry and Applications, Ministry of Education, Huaibei Normal University, Huaibei, P. R. China. ✉e-mail: hanmin@ustc.edu.cn

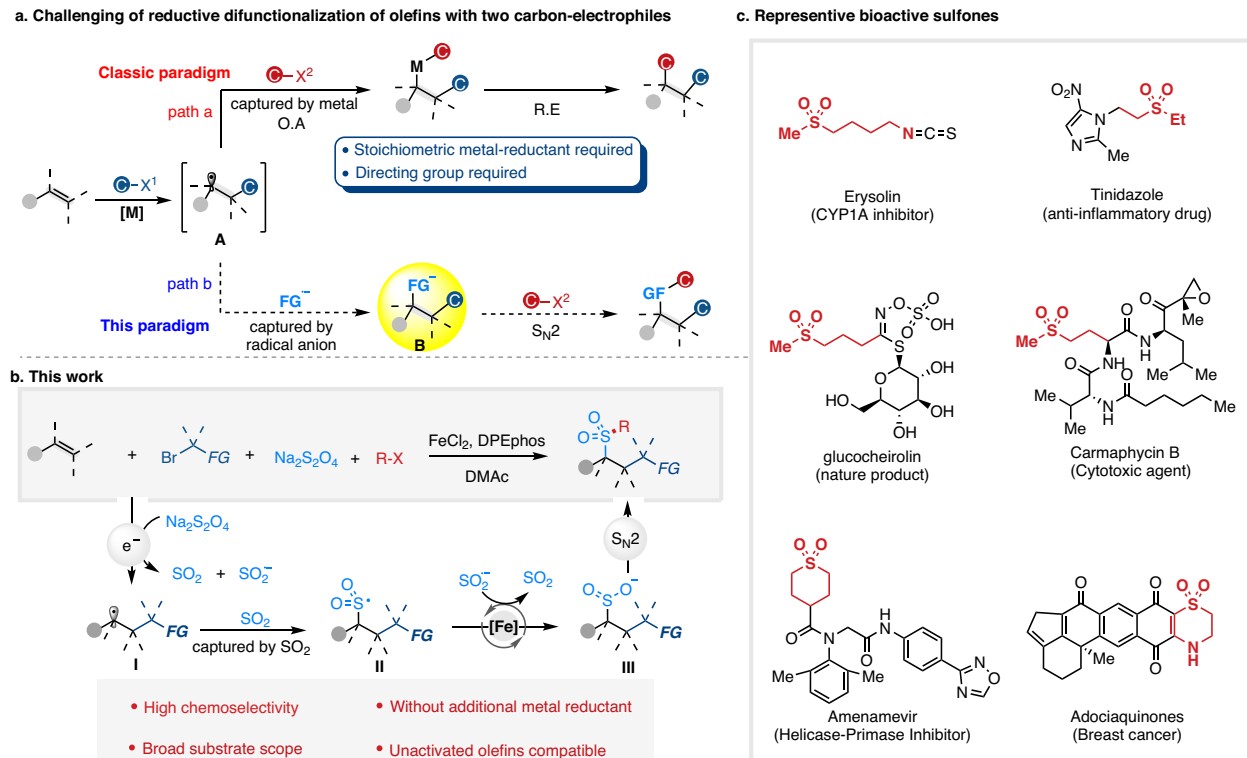

**Fig. 1 | Metal-catalyzed difunctionalization of alkenes. a** Comparison of catalytic cycles in traditional paradigm and our strategy. **b** Alkylative alkysulfonylation of alkenes. **c** Representative bioactive sulfones. M metal, R substituent, X leaving group, FG functional group, DPEphos 1-(diphenylphosphino)-2-(2-(diphenylphosphino)phenoxy)benzene.

studies have demonstrated $Na_2S_2O_4$ as an efficient single-electron reductant to convert the electron-deficient halogenated substances to the corresponding carbon-centered radicals[29,30]. Inspired by these results and based on our interests in electron-shuttle catalysis[31], we were cognizant of the fact that $Na_2S_2O_4$ could donate one electron to redox-active electron-deficient alkyl bromides, resulting in the releasing of $SO_2$ and generating an electron-deficient carbon-centered radical, which could be trapped by alkene to afford a novel radical intermediate **I**. The intermediate **I** was then captured by $SO_2$ to produce sulfonyl radical **II**, which can be smoothly reduced by a suitable metal-catalyst to produce alkylsulfonic anion **III**. The intermediate **III** is nucleophilic enough to react with other alkyl halides via nucleophilic substitution to furnish the desired difunctionalized products via the radical-anion relay. Meanwhile, the metal in a higher oxidation state could be reduced by another half of $Na_2S_2O_4$ to furnish the metal catalytic cycle (Fig. 1b). As such, the β-H elimination that existed in these reactions would be bypassed and the stoichiometric amounts of metal-reductants could be also omitted. Herein, we report a strategy to capture the alkylated carbon-centered radical **A** by $SO_2$, triggering an iron electron-shuttle-catalyzed multicomponent fluoroalkylative alkylsulfonylation of olefins[32–35]. A series of acyclic and cyclic sulfones were efficiently obtained with high selectivity, which were privileged scaffolds in material science, medicinal chemistry and nature products (Fig. 1c)[36–39].

## Results
### Reaction optimization
To validate the aforementioned hypothesis, a four-component coupling reaction of the readily available 1-octene (**S1-1**), 2-bromo −2,2-difluoro-*N*-phenylacetamide (BrCF$_2$CONHPh) (**S2-1**), bromoethane (EtBr) (**S3-1**) and $Na_2S_2O_4$ were examined (Fig. 2). The reaction was conducted at 100 °C with *N,N*-dimethylacetamide (DMAc) as solvent. To our delight, the desired product **1** could be

obtained in 26% yield, together with the non-alkene incorporated byproduct **1a** in 9% yield and 22% yield of mono-functionalized byproduct **1b** in the absence of catalyst, additive, and ligand (Fig. 2, entry 1). To improve the chemoselectivity of this transformation, various metal catalysts that are commonly utilized in radical-involved coupling reactions were investigated (Fig. 2, entries 2–6). Among the metal-catalyst screened, the iron-salts stood out as the effective catalysts as the monofunctionalized byproduct **1b** could be completely inhibited and the yield of the desired product **1** was improved to 34% when FeCl$_2$ acted as the catalyst. With FeCl$_2$ as the catalyst precursor, other reaction parameters were screened to maximize the efficiency of this coupling reaction (Fig. 2, entries 7–10). This study led to find that the use of FeCl$_2$ (10 mol%) as the catalyst and 1-(diphenylphosphino)−2-(2-(diphenylphosphino)phenoxy)benzene (DPEphos) (12 mol%) as the ligand gave the desired product **1** in 67% yield with high chemoselectivity. These results suggested that iron-catalyst play an important role to promote the $Na_2S_2O_4$ react with the in-situ formed carbon-centered radical. At the last stage of the optimization, the ratio of starting materials were fine-tuned and 81% isolated yield of the desired product was obtained when the loading of 2-bromo-2,2-difluoro-*N*-phenylacetamideand (**S2-1**) was increased to 1.2 equivalent (Fig. 2, entry 11). To further explore the role of iron, we synthesized the iron complex Fe(DPEphos)Cl$_2$, the solid-state structure of Fe(DPEphos)Cl$_2$ was unambiguously determined by single-crystal X-ray crystallographic analysis. Almost the same yield of the desired product **1** was obtained when the Fe(DPEphos)Cl$_2$ was used as the catalyst (Fig. 2, entry 12). Interestingly, when the ferrocene (Cp$_2$Fe) was used as the catalyst, 67% isolated yield of the desired product **1** was obtained (Fig. 2, entry 13). These results suggested that the metal-substrate coordination was most not likely involved in the present reaction, indicating that the iron-complex may act as an electron-shuttle to promote the desired reaction[40].

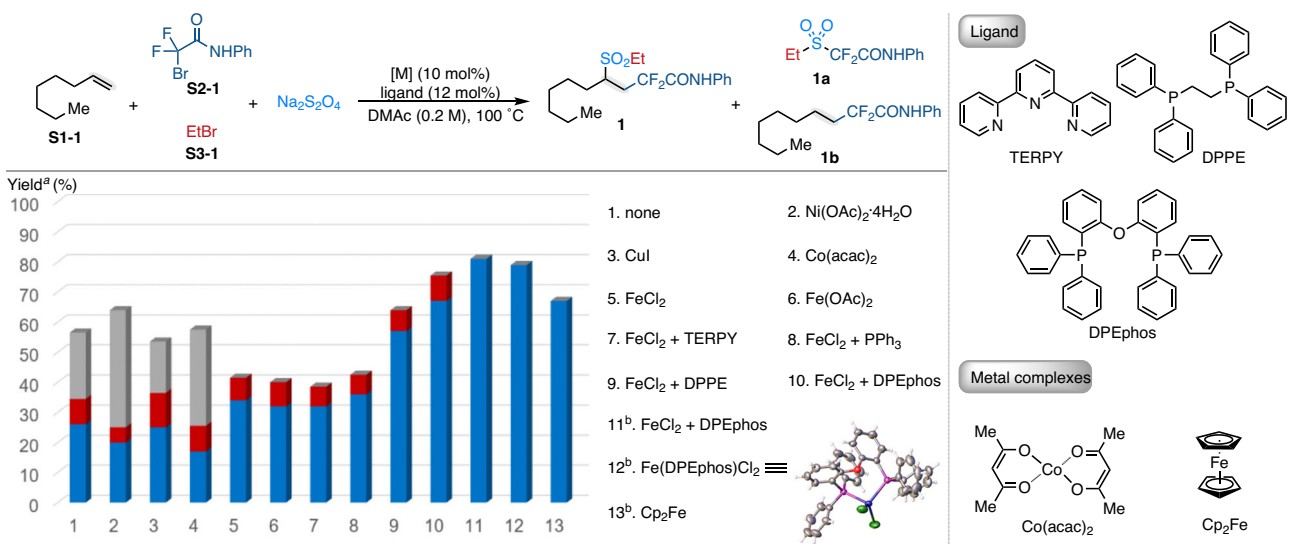

**Fig. 2 | Evaluation of the reaction conditions.** [a]Conditions: **S1-1** (0.2 mmol, 1.0 equiv.), **S2-1** (0.2 mmol, 1.0 equiv.), Na₂S₂O₄ (0.3 mmol, 1.5 equiv.), **S3-1** (0.3 mmol, 1.5 equiv.), [M] (10 mol%), ligand (12 mol%) in solvent (1.0 mL), 100 °C, 12 h; yields were determined by ¹H NMR analysis with dibromomethane as an internal standard. [b]**S2-1** (0.24 mmol, 1.2 equiv.); isolated yield. Blue cubes, **1**; red cubes, **1a**; gray cubes, **1b**.

## Scope of the methodology

With the optimal conditions established, the generality of this reaction was then investigated. First, a series of olefins were evaluated. As shown in Fig. 3, simple terminal alkenes, including 1-octenes and an array of nonfunctionalized alkenes with large steric hindrance, were amenable to this protocol to give the desired products (**1–4**, 60–87%) in good yields. Moreover, a series of unactivated olefins bearing different functional-groups, including alkene, ester, ketone, trimethylsilane (TMS), hydroxyl, phenyl-ring, and heteroaryl-rings were well tolerated in this transformation, affording the desired products (**5–17**, 67–93%) in excellent yields. However, aryl alkenes were not well adapted to this reaction, only styrenes containing electron-rich substituents can give the corresponding products in lower yields (**18–20**). Other activated olefins, bearing ether, ester, amide, and amine groups could be tolerated to give the corresponding products (**21–25**) in moderate yields (54–67%). We were also pleased to observe that this strategy was amenable to conjugated dienes, as the reaction of 1,3-butadiene furnished the 1,4-addition product (**26**) in moderate yield. Notably, this transformation proved to be capable of accommodating poly-substituted alkenes which gave rise to the products with quaternary carbon centers. For example, the 1,1-disubstituted ethylenes and exocyclic olefins exhibited excellent regioselectivities to give the desired products (**27–35**) in moderate to excellent yields (53%-91%). The solid-state structure of **27** was unambiguously determined by single-crystal X-ray crystallographic analysis. Symmetric 1,2-disubstituted olefins were also well compatible to give the corresponding products **36** and **37** in moderate yields. Furthermore, when the cyclic alkenes with five- to eight-membered rings were used as substrates, the desired products could also be obtained (**38–42**, 46–91%). Thereinto, 2,5-dihydrofuran, cyclohexene, and alky-substituted cyclohexenes selectively deliver the products with trans-configurations (**38–41**, >20:1 dr). To further determine the configuration of the products, the solid-state structure of **38** and **40** were unambiguously determined by single-crystal X-ray crystallographic analysis. Then, the trisubstituted and tetrasubstituted alkenes gave the target products (**43–45**, 33–88%) with good regioselectivities and moderate to good yields. Interestingly, 2,2-difluoro-4-(methylsulfonyl)-N-phenylbutanamide (**46**) could be obtained in 57% yield, when the trimethyl(vinyl)silane served as alkene. The trimethyl(vinyl)silane could be used to replace ethylene gas to obtain the corresponding product. To our delight, this transformation was not limited to simple olefins. Alkenes decorated with extensively existing natural product skeletons and pharmacophores were well applicable to the present reaction (**47–53**). For example, the commercially available nootkatone bearing a disubstituted double bond was successfully engaged in the standard conditions to give the desired product **53** in 63% yield.

Next, we evaluated the scope of the two kinds of alkyl halides, which revealed a range of radical precursors could be employed. Firstly, we test the scope of radical precursors by using 2-(but-3-en-1-yl)naphthalene as the alkene component and EtBr as the alkyl bromide component. As shown in Fig. 4, a series of aliphatic bromodifluroacetamides was well tolerated in this catalytic paradigm and gave desired products (**54–56**, 52–64%) in moderate yields. In addition, other electron-deficient radical precursors, including ethyl bromodifluoroacetate bromodifluoromethylsulfone, and bromodifluoromethylbenzoxazole were successfully assembled in this transformation, providing the desired products (**57–59**, 46–65%) in good yields. This catalytic paradigm was also successfully applied to 1,2-dibromo-1,1,2,2-tetrafluoroethane and perfluorobutyl iodide, providing corresponding fluoroalkylated sulfones (**60**, **61**, 42–61%) in moderate yields. Less electron-deficient radical precursors, such as 2-bromo-2,2-difluoroethan-1-ol and nonfluorinated bromoacetate, could be successfully adapted to this protocol, gave the desired products (**62**, **63**, 47–49%) in moderate yields. Moreover, the L-tryptophan and aspartame-derived bromodifluoroacetamides could be smoothly converted to fluorine-decorated products in good yields (**64**, **65**, 67%, and 66%, respectively).

With 2-methylprop-1-ene as the alkene coupling partner and BrCF₂CONHPh as the radical precursor, a series of simple alkyl electrophiles were evaluated. Leaving groups including −I, −Br, −Cl, −OTs, and −OMs were all compatible with the catalytic paradigm to deliver product **66** in good yields. Functional-groups decorated to the primary alkyl bromides, such as cyclopropyl, alkenyl, alkynyl, silyl ether, ketone, acetal, ester, aryl, nitrile, and amide, were all well-tolerated (**67–79**, 41–89% yield). Notably, the relatively more active benzyl bromide and 1-(bromomethyl)naphthalene were also tolerant. Besides, a series of secondary haloalkanes including cyclic, non-cyclic, and functional halogenates were well tolerated, offering compounds (**80–83**, 42–92%) in moderate to excellent yields. Moreover, 1-chloro-4-nitrobenzene proved to be a suitable substrate to give the desired product **84** in moderate yield. In addition, the protocol could be also

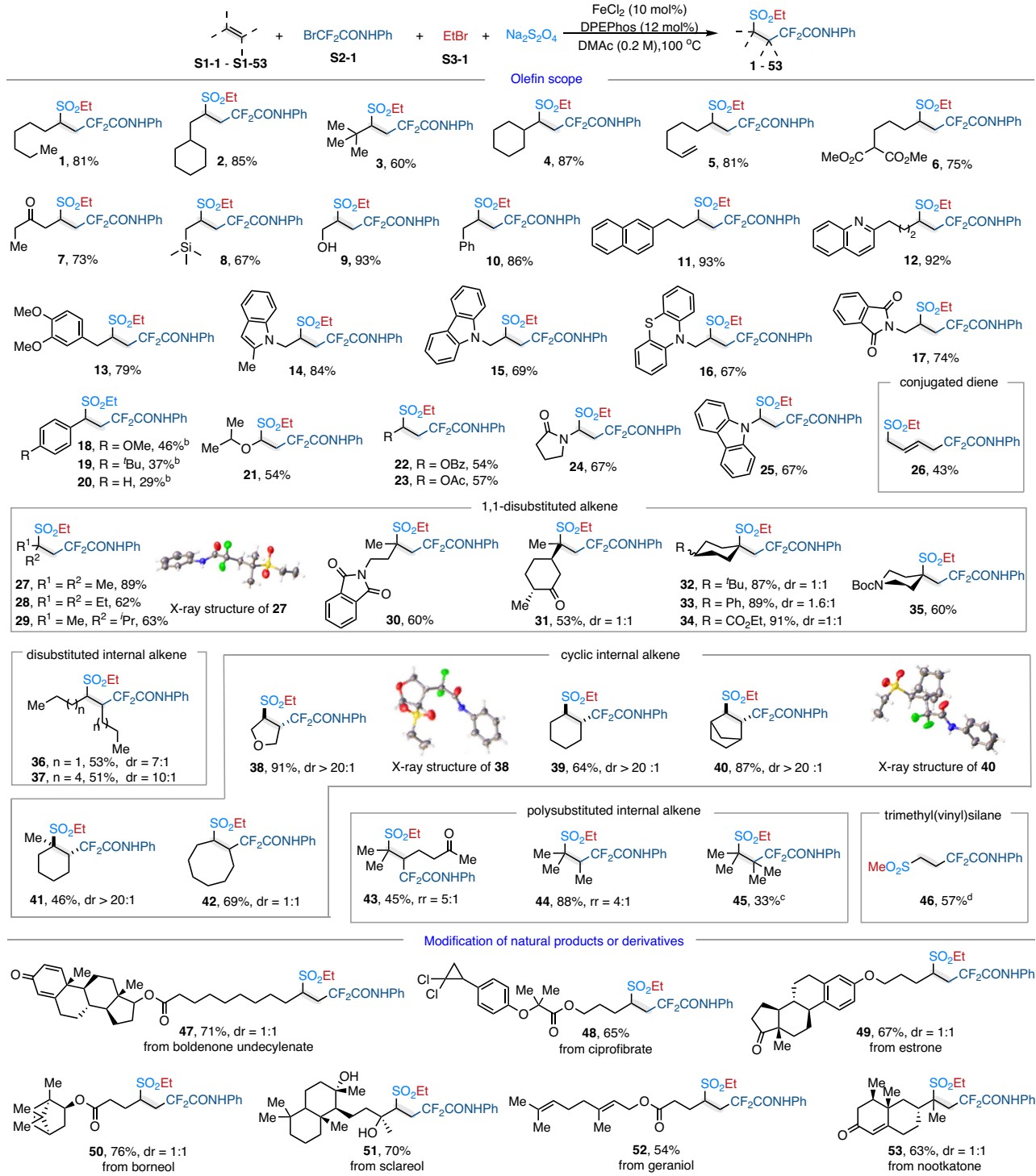

**Fig. 3 | Olefin scope.** [a]Conditions: alkene **S1** (0.4 mmol, 1.0 equiv.), radical precursor **S2-1** (0.48 mmol, 1.2 equiv.), alkylbromide **S3-1** (0.6 mmol, 1.5 equiv.), FeCl₂ (10 mol%), DPEPhos (12 mol%), Na₂S₂O₄ (0.6 mmol, 1.5 equiv.) in DMAc (2.0 mL), 100 °C, 12 h, under nitrogen atmosphere. Isolated yields after chromatography are given. [b]The reaction temperature was conducted at 70 °C. [c]Olefin (0.8 mmol, 2.0 equiv.), bromoethane (0.4 mmol, 1.0 equiv.). [d]**S3** was MeOTs; Ligand was 2,2':6',2"-terpyridine. Ac acetyl, Bz benzoyl, Boc tert-butyloxycarbonyl, Me methyl, Et ethyl, DPEphos 1-(diphenylphosphino)-2-(2-(diphenylphosphino)phenoxy)benzene.

utilized to synthesize annulated-sulfones, which are the most important motifs in pharmaceuticals[41]. By using alkene-tethered alkyl bromides as the starting materials, a variety of cyclic sulfones bearing four- to eight-membered rings were systematically prepared in moderate to excellent yields (**85**–**96**, 59–92%), although some *O*-alkylated cyclization byproducts were detected under this transformation. By prolonging the tether-length, the 14-membered cyclic-sulfone (**97**) was also obtained with promising efficiency.

## Synthetic applications

A gram-scale reaction and the functional group transformation of products were conducted to further demonstrate the synthetic versatility of this catalytic transformation. Firstly, 1.21 g desired hydroxyl-containing product **9** was obtained with 4 mmol allyl alcohol as the starting materials under the standard conditions (Fig. 5a). Then, a series of transformations of product **9** were established. As shown in Fig. 5a, the hydroxyl-featured sulfone **9** could be smoothly

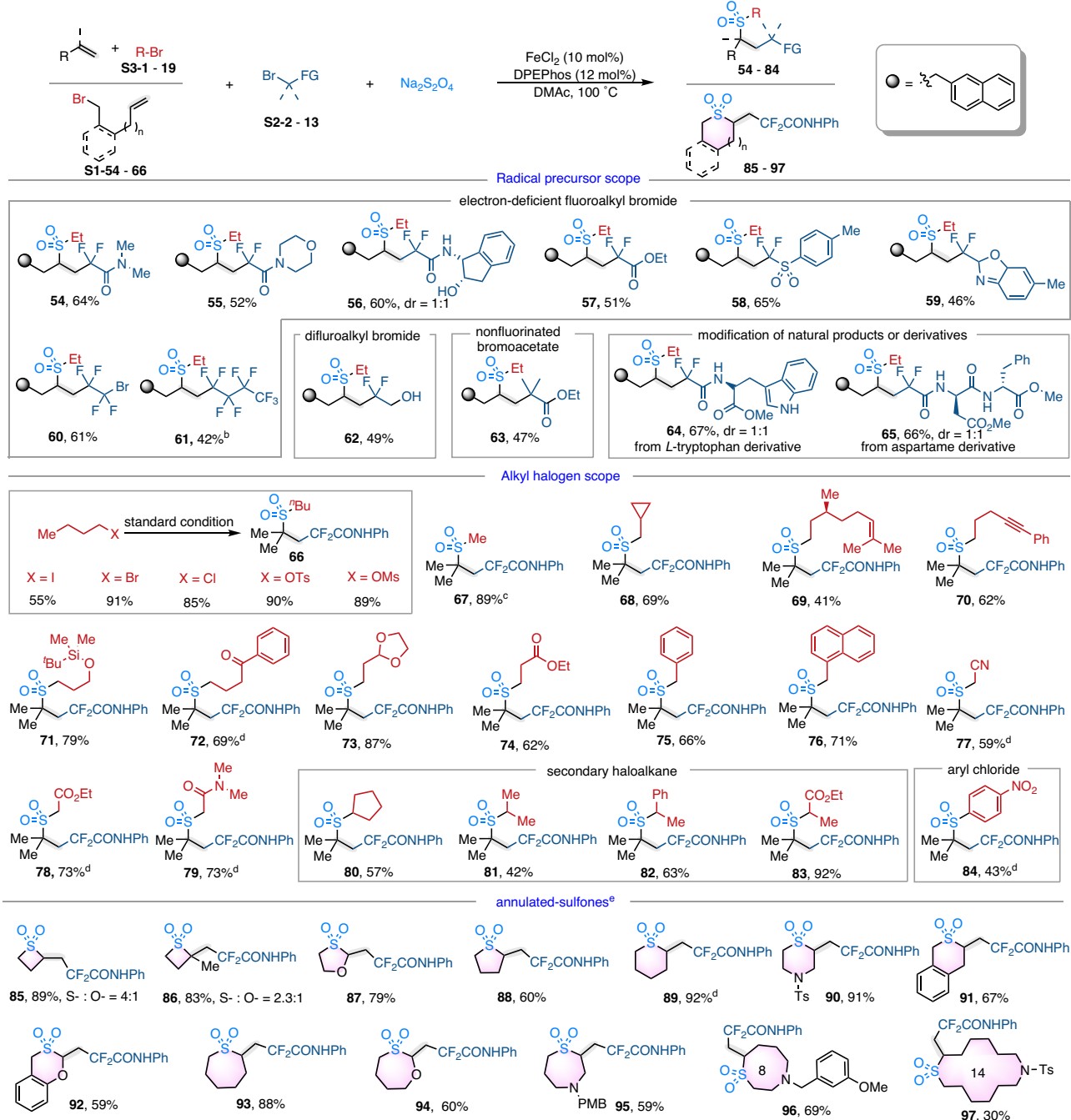

**Fig. 4 | Electrophiles scope.** [a]Condition: alkene **S1** (0.4 mmol, 1.0 equiv.), **S2** (0.48 mmol, 1.2 equiv.), **S3** (0.6 mmol, 1.5 equiv.), FeCl$_2$ (10 mol%), DPEphos (12 mol%), Na$_2$S$_2$O$_4$ (0.6 mmol, 1.5 equiv.) in DMAc (2.0 mL), 100 °C, 12 h under nitrogen; isolated yields are given. [b]The electrophile was iodoalkane. [c]The electrophile was Me-OTs. [d]The electrophile was chlorinated alkane. [e]DMAc (4.0 mL). Ph phenyl, PMB *p*-methoxybenzyl, Me methyl, Et ethyl, [n]Bu *n*-butyl, DPEphos 1-(diphenylphosphino)-2-(2-(diphenylphosphino)phenoxy)benzene, [t]Bu *t*-butyl, Ts tosyl. Gray ball, 2-methylenenaphthalene.

converted to alkyl silyl ethers **98** in 89% yield by hydroxyl-protection. In addition, the α,β-unsaturated sulfone **99** was efficiently obtained through treatment with methanesulfonyl chloride (MsCl) and *N,N*-diisopropylethylamine (DIPEA). Furthermore, product **99** could be further converted into chained polyfluorinated sulfone-containing compounds **100** with Na$_2$S$_2$O$_4$ as the radical initiator. Next, the amide-featured sulfone (**9**) could be smoothly and selectively converted to amino alcohol (**101**) in excellent yield through reduction with BH$_3$-THF. Using CBr$_4$ as the brominated reagent, piperidine derivatives **102** could be obtained in one step. The enylamine (**103**) could be afforded in excellent yield by treatment of **100** with MsCl

and *N,N*-diisopropylethylamine (DIPEA). To further validate the synthetic value of the present method, the intermediates for the synthesis of Erysolin and its fluorinated derivatives have been prepared (Erysolin is an important CYP1A inhibitor with antigenic toxicity). As shown in Fig. 5b, the difunctionalization product **104** was obtained in 61% yield by using trimethyl(vinyl)silane as the alkene coupling partner. The sulfone **104** could be converted to the defluorination product **107** with SmI$_2$/Et$_3$N/H$_2$O[42], which could be further used to synthesize converted the desired Erysolin according to the known method[43,44]. In addition, the difluorinated Erysolin-2F could be also synthesized from the compound **105**, which was

**Fig. 5 | Investigation of the utility of this transformation. a** Gram-scale synthesis. **b** Post-functionalization of product. Conditions: (a) TBSCl (1.0 equiv.), imidazole (1.5 equiv.), DMF/DCM = 1/9 (v/v), 0 °C-rt, 16 h; (b) MsCl (1.5 equiv.), DIPEA (1.5 equiv.), DCM (0.5 M), rt, 2 h; (c) Na$_2$S$_2$O$_4$ (2.0 equiv.), DMAc (0.2 M), 100 °C, 10 h; (d) BH$_3$-THF (2.0 equiv.), THF (0.2 M), 60 °C, 24 h; (e) CBr$_4$ (1.3 equiv.), PPh$_3$ (1.3 equiv.), DCM (0.5 M), 0 °C-rt, 8 h; (f) MsCl (1.5 equiv.), DIPEA (1.5 equiv.), DCM (0.5 M), rt, 2 h. (g) NEt$_3$ (15 equiv.), SmI$_2$ (8.0 equiv.), H$_2$O (23 equiv.), THF, rt, 15 min. (h) NEt$_3$ (15 equiv.), SmI$_2$ (8.0 equiv.), H$_2$O (23 equiv.), THF, rt, 12 h. Et ethyl, Ph phenyl, TBS *tert*-Butyldimethylsilyl, PMP 4-methoxyphenyl.

produced by reduction of the sulfone **104** with BH$_3$-THF. Moreover, the monofluorinated Erysolin-F might be prepared from monofluorinated sulfone **106** by the same strategy.

## Mechanistic investigations

Based on previous reports in the field of metal electron-shuttle catalysis and Na$_2$S$_2$O$_4$-involved coupling reactions, a plausible radical-anion relay pathway is suggested in Fig. 6a. First, the electron-deficient alkyl bromide accepts an electron from Na$_2$S$_2$O$_4$ to generate carbon radical **A** together with the release of SO$_2$[34]. The radical-olefin addition reaction takes place to form the carbon-centered radical **B**, which is subsequently captured by SO$_2$ and produces sulfonyl-radical **C**[45]. The intermediate **C** is reduced by Fe$^{II}$ to generate Fe$^{III}$ species and sulfone anion **D**[46], which is further captured by another electrophile to furnish the desired product via nucleophilic substitution. Eventually, the Fe$^{III}$ is reduced by another half of Na$_2$S$_2$O$_4$ to regenerate Fe$^{II}$, furnishing the iron electron-shuttle catalytic cycle. Notably, the conversion of carbon radical **B** to sulfone radical **C** is reversible[47,48]. Intermediate **B** would accumulate if the efficient reductant was absent for the reduction of sulfone radical **C**, which results in the formation of by-product **1b**.

To support the hypothesis, we conducted a number of control experiments. First, the radical-clock experiments were conducted and revealed that alkyl radical intermediates **A** and **B** were most likely involved in this transformation. However, when the (bromomethyl) cyclopropane was used as a coupling partner, the desired product **68** was obtained in 69% yield and no ring-opening byproduct **68'** was detected, which agreed with our proposal that the final step took place via nucleophilic substitution (Fig. 6b). When the standard reaction was conducted in the presence of a stoichiometric amount of DABSO/FeCl$_2$/DPEphos or FeCl$_3$/DPEphos, the reaction was completely inhibited. On the other hand, the desired product **1** could be obtained in 37% yield, when a catalytic amount of FeCl$_3$ was added. These results indicated that the reaction was most likely initiated by the Na$_2$S$_2$O$_4$, and a stoichiometric amount of Fe$^{III}$ can react with Na$_2$S$_2$O$_4$ to inhibit the reaction (Fig. 6c). Further control experiments confirmed that SO$_2$ was involved in the reaction to capture the radical **B**, as the yield of the desired product **25** was dramatically decreased when the reaction was purged with nitrogen gas (Fig. 6d) (see Supplementary Figs. 4 and 5 for details). To improve our understanding of the reaction, we further carried out a series of time course experiments, which monitored the conversion of BrCF$_2$CONHPh and yields of the desired product **1** and byproduct **1b** over time either under the catalysis of Fe-catalyst or without catalyst. As shown in Fig. 6e, compared to the non-catalytic system, the iron-catalyst significantly decreased the consumption-rate of the

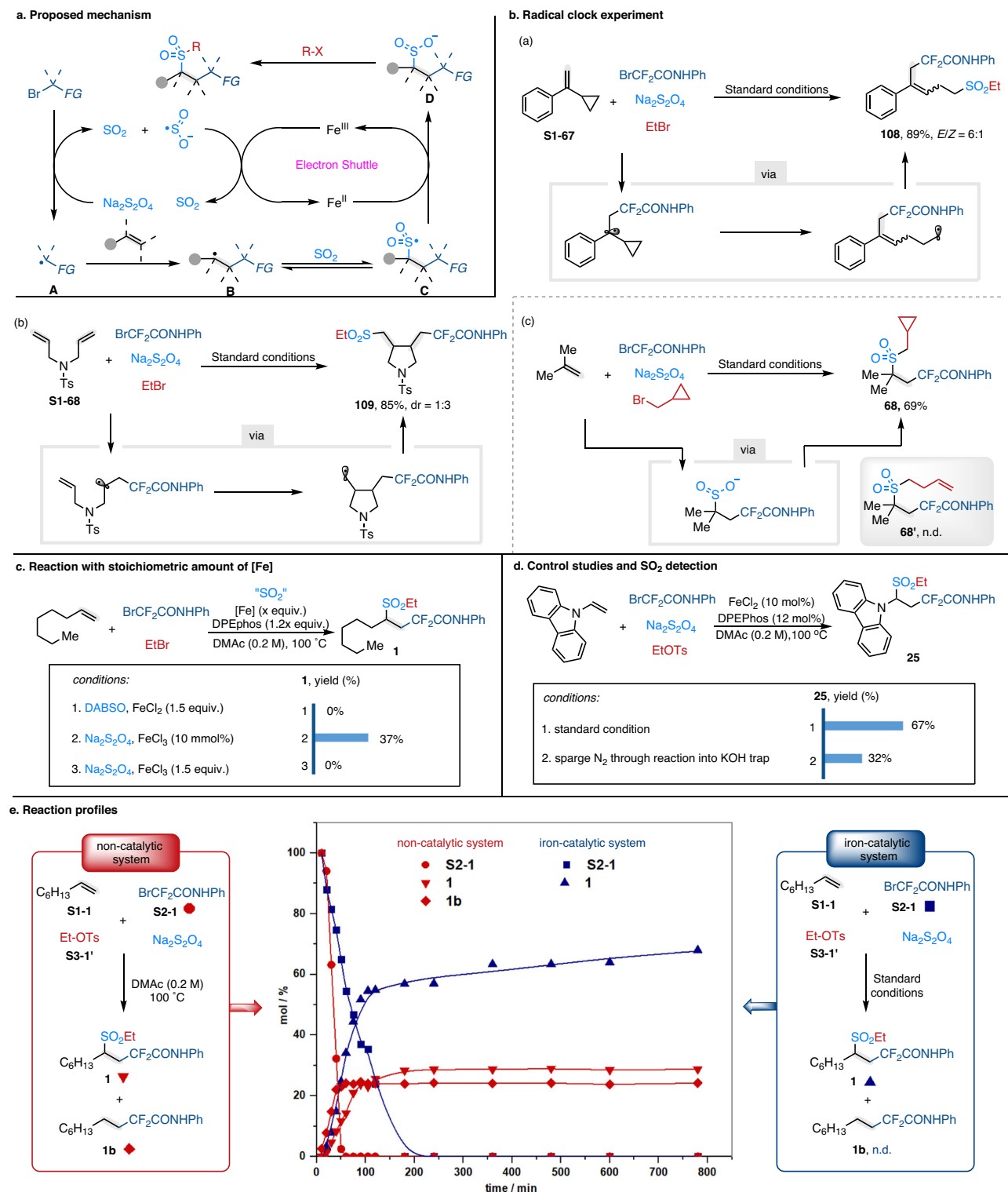

**Fig. 6 | Mechanistic investigation. a** Proposed mechanism. **b** Radical clock experiments. **c** Reaction with stoichiometric amount of [Fe]. **d** Control studies and SO₂ detection. **e** Reaction profiles. For detailed conditions, see the Supplementary Information. Me methyl, Et ethyl, Ts tosyl, Ph phenyl, n.d. not detected. Non-catalytic system: red ball, **S2-1**; red triangle, **1**; red diamond, **1b**. Catalytic system: blue square, **S2-1**; blue triangle, **1**.

BrCF₂CONHPh and completely inhibited the formation of non-sulfonylated byproduct **1b**. These results further confirmed that the reaction was initiated by the Na₂S₂O₄ and Feᴵᴵ acted as electron-shuttle to convert the sulfonyl-radical **C** to the corresponding anion **D**. The in-situ formed Feᴵᴵᴵ would react with the dissolved Na₂S₂O₄, thus decreasing the consumption rate of the BrCF₂CONHPh.

## Discussion

In summary, the elusive difunctionalization of alkenes with two different alkyl halides and forging C-C and C-S bonds has been developed by taking advantage of the radical involved electron-shuttle catalysis mechanism. The use of Na₂S₂O₄ was found to be crucial to divert the mechanistic pathway toward the formation of alkyl sulfone

anion intermediate instead of the previously reported alkylmetal intermediates, which successfully suppressed the β-H elimination existed in typical alkene-difunctionalization reactions. Mechanistic studies allowed the delineation of a mechanistic pathway including carbon-centered alkyl radical and sulfur-centered sulfone radical, which are in equilibrium via capture or extrusion of $SO_2$. With Fe(D-PEphos)$Cl_2$ or $Cp_2Fe$ as a catalyst, reducing the sulfone radical to sulfone anion is favored, leading to an efficient fluoroalkylative alkylsulfonylation reaction with high chemo- and regioselectivity. This multicomponent reaction proceeded via radical-anion relay tolerates a variety of functional groups and showed a broad scope for the generation of diverse dialkyl sulfones, including valuable cyclic sulfones, from simple alkenes and alkyl halides. This work further illustrated that tuning the catalytic paradigm in metal-catalyzed reductive alkene-difunctionalization reactions may allow to divert mechanistic pathways toward the formation of high value-added products.

## Methods

### General procedure for the four-component reaction

FeCl$_2$ (5.0 mg, 10 mol%), DPEphos (25.9 mg, 12 mol%), olefins (**S1**, 0.40 mmol, 1.0 equiv.), radical precursor (**S2**, 0.48 mmol, 1.2 equiv.), alkyl bromide (**S3**, 0.60 mmol, 1.5 equiv.), $Na_2S_2O_4$ (104.4 mg, 0.60 mmol, 1.5 equiv.) and DMAc (2.0 mL) were added to a flame-dried Young-type tube under $N_2$ atmosphere. The reaction mixture was stirred (770 rpm) at 100 °C for 12 h. After completion, the reaction was cooled to room temperature. The reaction mixture was diluted with $H_2O$. Then aqueous phase was extracted with EtOAc (3 × 10 mL). The combined organic extracts were washed with brine, dried over $Na_2SO_4$, filtered, and concentrated in vacuo. The residue was purified by silica gel column chromatography to afford the desired product.

### General procedure for the three-component reaction

FeCl$_2$ (5.0 mg, 10 mol%), DPEphos (25.9 mg, 12 mol%), 2-bromo-2,2-difluoro-$N$-phenylacetamide **S2-1** (120.0 mg, 0.48 mmol, 1.2 equiv.), alkene-tethered alkyl bromides (**S1**, 0.4 mmol, 1.0 equiv.), $Na_2S_2O_4$ (104.4 mg, 0.60 mmol, 1.5 equiv.) and DMAc (4.0 mL) were added to a flame-dried Young-type tube under $N_2$ atmosphere. The reaction mixture was stirred (1000 rpm) at 100 °C for 12 h. After completion, the reaction was cooled to room temperature. The reaction mixture was diluted with $H_2O$. Then aqueous phase was extracted with EtOAc (3 × 20 mL). The combined organic extracts were washed with brine, dried over $Na_2SO_4$, filtered, and concentrated in vacuo. The residue was purified by silica gel column chromatography to afford the desired product.

## Data availability

All data supporting the findings of this study are available within the article and its Supplementary Information files or from the corresponding author upon request. Crystallographic data for the structures reported in this article have been deposited at the Cambridge Crystallographic Data Centre, under deposition numbers CCDC 2294876 (**27**), 2294877 (**38**), 2294886 (**40**) and 2294887 (Fe(DPEphos)Cl$_2$). Copies of the data can be obtained free of charge via https://www.ccdc.cam.ac.uk/structures/.

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

## Acknowledgements

Financial support for this project was provided by the National Natural Science Foundation of China (21925111, 22301290, 22350008, 22301289 and 92356302), the Strategic Priority Research Program of Chinese Academy of Sciences (XDB0450301), and National Key R&D Program of China (2021YFA1501003) (H.H.). This work was partially carried out at the Instruments Center for Physical Science, University of Science and Technology of China.

## Author contributions

H.H. conceived the concept and directed the project. X.H. and H.L. conducted the experiments. X.H. and H.H. wrote the paper. All the authors discussed and analyzed the results and commented on the manuscript.

## Competing interests

The authors declare no competing interests.
