## [Peer Review File · Nature Communications]

Iron-catalyzed alkylative alkylsulfonylation of alkenes via radical-anion relayREVIEWER COMMENTS

Reviewer #1 (Remarks to the Author):

In this manuscript, the authors report a Fe-catalyzed difunctionalization of alkenes with Na₂S₂O₄ serving as both reductant and sulfone-source. The use of Na₂S₂O₄ avoids excess amounts of metallic reductant and provides a new strategy to install sulfone groups. The reaction exhibits good functional group tolerance and wide substrate scope including both activated and non-activated alkenes, as well as different radical precursors. This work provides a good contribution to the field of alkene difunctionalization, which I believe has considerable interest for the readers of Nat. Commun. However, some issues need to be addressed before the manuscript can be considered for publication:

1. Examples of bioactive sulfones are provided in Fig 1. Can the authors show that their method allows access to some these drug molecules or intermediates?
2. Will the reaction take place on conjugated alkenes (e.g. 1,3-dienes)? If so, what is the regiochemical outcome?

Reviewer #2 (Remarks to the Author):

In this manuscript, Huang and co-workers describe an iron-catalyzed difunctionalization of alkenes with two different alkyl halides via a radical-anion relay strategy. By using Na₂S₂O₄ as both reductant and sulfone-source, this reaction provide a facile method for preparing various dialkyl sulfones with high regio- and chemoselectivity. The authors made much efforts to expand the substrate scope with respect to alkenes and alkyl halides. Control experiments are well performed to illustrate the reaction mechanism.

However, the main content of this protocol is based on the widely explored SO₂ insertion strategy. Moreover, Na₂S₂O₄ serving as the dual role of reductant and sulfone-source has been demonstrated by other groups as well (From Scifinder, a review is appeared from Wu's group: Chin. Chem. Lett. 2021, 32, 461-464.). The main highlight of this manuscript is switching to alkene difunctionalization for the generation of new carbon-centered radicals to be captured by SO₂. It seems that this work is an extension of the previous reports in this field, no matter from the reaction design or the reaction mechanism. In my opinion, this work is not novelty and attractive enough to be published on Nat. Commun. Publication in in a more specialized journal is suggested.

Reviewer #3 (Remarks to the Author):

In this manuscript, Huang and co-workers report an unprecedented iron-catalyzed four-component difunctionalization of alkenes to install alkylsulfonyl and fluoroalkyl groups across the double bond in a regioselective fashion. This transformation delivers, in moderate to high yields, a wide array of densely functionalized compounds. This method is operationally simple, exhibits a remarkable and large functional group tolerance and offer a new strategy to access fluorine-containing frameworks of high interest. In addition, the authors demonstrate the important synthetic utility of these compounds through a variety of derivatizations, present a solid mechanistic study that globally support their mechanistic proposal along a well-described the supporting information. Overall, I think that it is an excellent paper that is suitable for publication in NatureComm. However, several points need

to be addressed first:

(1) The title should be modified as it is misleading, the authors used the term “alkylative” while “fluoroalkylative” would be more adapted (a single example without fluorine groups installed is presented).

(2) Key references are missing, notably Chem. Eur. J. 2019, 25, 1824 that displays relatively similar frameworks while less complex than in the present paper. It should be clearly mentioned in the introduction as prior art. The following references should be added too: Green Chem 2023, 25, 9292; Org. Lett. 2020, 22, 2801; Chem 2021, 7, 3412.

(3) In Scheme 1, the authors present several bioactive molecules incorporating a sulfone motif. Yet, not a single of them closely resemble the compounds prepared in the present paper. Thus, either the authors remove them since they are not relevant to this study or the authors present molecules that can at least vaguely be linked to their products.

(4) In terms of scope, it seems that styrenes are a limitation. However, this point is not commented by the authors as they only show a single example. They should give a clearer picture of this reactivity by reporting styrenes incorporating strong electron-withdrawing groups (CF₃ for instance) and electron-donating groups (OMe for instance), the simple styrene and a vinyl heteroaromatic.

(5) A question of curiosity: could aryl bromides instead of alkyl ones could be used to generate aryl sulfones?

(6) Regarding the mechanism, the authors concluded that the last step occurred via a nucleophilic substitution because the bromomethylcyclopropane does not open. However, this compound is not fully activated, does they get the same result with a phenyl-substituted one?

(7) I do not know if it is due to my computer, but NMR spectra appear a bit blurry and are not always easy to look at. Please fix this issue.

Reply to Reviewer 1 comments:

Reviewer 1 appropriately evaluates our results. I am grateful to **Reviewer 1**, who gave us valuable suggestions to support and improve our manuscript.

In this manuscript, the authors report a Fe-catalyzed difunctionalization of alkenes with Na₂S₂O₄ serving as both reductant and sulfone-source. The use of Na₂S₂O₄ avoids excess amounts of metallic reductant and provides a new strategy to install sulfone groups. The reaction exhibits good functional group tolerance and wide substrate scope including both activated and non-activated alkenes, as well as different radical precursors. This work provides a good contribution to the field of alkene difunctionalization, which I believe has considerable interest for the readers of Nat. Commun. However, some issues need to be addressed before the manuscript can be considered for publication:

(1) Examples of bioactive sulfones are provided in Fig 1. Can the authors show that their method allows access to some these drug molecules or intermediates?

Reply: Thanks for your comments. We have tried our best to synthesize some bioactive molecules shown in the Fig. 1c by using our newly developed protocol as a key reaction step, but failed. However, we can successfully prepared the compound **107** by using our catalytic reaction as a key reaction step. The compound **107** could be utilized as a key intermediate to prepare the Erysolin, which is an important CYP1A inhibitor with antigenic toxicity (*ChemBioChem* **2008**, *9*, 729–747; *J. Asian Nat. Prod. Res.* **2023**, *25*, 369–378). Moreover, the fluorinated analogues **105** and **106** could be also prepared by using this method, which could be utilized to synthesize the fluorinated Erysolins. On the other hand, cyclic sulfones are frequently found in the bioactive molecules. As such, we have revised the Fig. 1c.

Fig. 1c

2,2-Difluoro-N-(4-methoxyphenyl)-4-(methylsulfonyl)butanamide (104)

$^1\text{H NMR}$ (400 MHz, $\text{DMSO-}d_6$) δ 10.55 (s, 1H), 7.58 (d, $J = 9.0$ Hz, 2H), 6.94 (d, $J = 9.0$ Hz, 2H), 3.74 (s, 3H), 3.41 – 3.29 (m, 2H), 3.09 (s, 3H), 2.70 – 2.54 (m, 2H). $^{13}\text{C NMR}$ (101 MHz, $\text{DMSO-}d_6$) δ 160.80, 156.44, 130.01, 122.58, 116.69, 113.92, 55.26, 46.52, 40.20, 27.10 (t, $^2J_{\text{C-F}} = 24.8$ Hz). $^{19}\text{F NMR}$ (376 MHz, $\text{DMSO-}d_6$) δ -104.10 (s, 2F). **HRMS** (ESI) calcd for $\text{C}_{12}\text{H}_{15}\text{F}_2\text{NNaO}_4\text{S}$ $[\text{M} + \text{Na}]^+$: 330.0582, found: 330.0588.

N-(2,2-Difluoro-4-(methylsulfonyl)butyl)-4-methoxyaniline (105)

$^1\text{H NMR}$ (400 MHz, CDCl_3) δ 6.80 (d, $J = 8.8$ Hz, 2H), 6.64 (d, $J = 8.8$ Hz, 2H), 3.75 (s, 3H), 3.64 (d, $J = 6.8$ Hz, 1H), 3.52 (td, $J = 12.8, 5.6$ Hz, 2H), 3.31 – 3.20 (m, 2H), 2.94 (s, 3H), 2.61 – 2.41 (m, 2H). $^{13}\text{C NMR}$ (126 MHz, CDCl_3) δ 153.21, 140.93, 122.60 (t, $^1J_{\text{C-F}} = 244.2$ Hz), 115.11, 114.88, 55.87, 50.14 (t, $^2J_{\text{C-F}} = 29.5$ Hz), 47.93 (t, $^3J_{\text{C-F}} = 4.1$ Hz), 41.02, 27.72 (t, $^2J_{\text{C-F}} = 25.1$ Hz). $^{19}\text{F NMR}$ (376 MHz, CDCl_3) δ -103.91 (s, 2F). **HRMS** (ESI) calcd for $\text{C}_{12}\text{H}_{18}\text{F}_2\text{NO}_3\text{S}$ $[\text{M} + \text{H}]^+$: 294.0970, found: 294.0979.

2-Fluoro-N-(4-methoxyphenyl)-4-(methylsulfonyl)butanamide (106)

¹H NMR (500 MHz, CDCl₃) δ 7.93 (s, 1H), 7.45 (d, *J* = 9.0 Hz, 2H), 6.89 (d, *J* = 9.0 Hz, 2H), 5.18 (ddd, *J* = 48.9, 7.1, 4.6 Hz, 1H), 3.81 (s, 3H), 3.45 – 3.08 (m, 2H), 2.95 (s, 3H), 2.86 – 2.42 (m, 2H). **¹³C NMR** (126 MHz, CDCl₃) δ 166.08 (d, ²*J*_{C-F} = 18.0 Hz), 157.30, 129.32, 122.07, 114.48, 89.57 (d, ¹*J*_{C-F} = 190.4 Hz), 55.64, 49.95 (d, ³*J*_{C-F} = 3.8 Hz), 40.83, 25.85 (d, ²*J*_{C-F} = 21.1 Hz). **¹⁹F NMR** (471 MHz, CDCl₃) δ -188.34 (s, 1F). **HRMS** (ESI) calcd for C₁₂H₁₆FNNaO₄S [M + Na]⁺: 312.0676, found: 312.0683.

***N*-(4-Methoxyphenyl)-4-(methylsulfonyl)butanamide (107)**

¹H NMR (500 MHz, CDCl₃) δ 7.40 (d, *J* = 9.0 Hz, 3H), 6.86 (d, *J* = 8.9 Hz, 2H), 3.79 (s, 3H), 3.17 (t, *J* = 7.2 Hz, 2H), 2.94 (s, 3H), 2.60 (t, *J* = 6.8 Hz, 2H), 2.28 (p, *J* = 7.0 Hz, 2H). **¹³C NMR** (126 MHz, CDCl₃) δ 169.47, 156.71, 130.78, 121.94, 114.33, 55.63, 53.42, 41.05, 34.78, 18.67. **HRMS** (ESI) calcd for C₁₂H₁₇NNaO₄S [M + Na]⁺: 294.0770, found: 294.0777.

¹³C NMR Spectra of 104

hxy-4-87-1.4.fid

¹⁹F NMR Spectra of 104

hxy-4-87-1.3.fid

¹H NMR Spectra of 105

hxy-4-88-2.10.fid

¹³C NMR Spectra of 105

HXY-4-88-2.1.fid

¹⁹F NMR Spectra of 105

hxy-4-88-2.11.fid

-103.91

¹H NMR Spectra of 106

¹³C NMR Spectra of 106

hxy-4-87-3-1.2.fid

¹⁹F NMR Spectra of 106

hxy-4-87-3-1.3.fid

¹H NMR Spectra of 107

hxy-4-87-3-2.1.fid

¹³C NMR Spectra of 107

hxy-4-87-3-2.2.fid

(2) Will the reaction take place on conjugated alkenes (e.g. 1,3-dienes)? If so, what is the regiochemical outcome?

Reply: When the 1,3-butadiene was used as the conjugated alkene, only 1,4-addition product **26** was obtained in 43% isolated yield. In this reaction, an allyl radical was generated, and the site-selectivity would mainly depend on the steric effect, thus the 1,4-addition became the predominant process in these reactions.

(E)-6-(Ethylsulfonyl)-2,2-difluoro-N-phenylhex-4-enamide (26)

The title compound was prepared according to the *General procedure I* and purified by column chromatography as a yellow solid (54.5 mg, 43% yield). $^1\text{H NMR}$ (400 MHz, CDCl_3) δ 8.08 (s, 1H), 7.75 – 7.53 (m, 2H), 7.46 – 7.31 (m, 2H), 7.25 – 7.16 (m, 1H), 5.98 – 5.73 (m, 2H), 3.69 (d, $J = 6.1$ Hz, 2H), 3.02 (td, $J = 16.3, 5.7$ Hz, 2H), 2.91 (q, $J = 7.5$ Hz, 2H), 1.29 (t, $J = 7.5$ Hz, 3H). $^{13}\text{C NMR}$ (101 MHz, CDCl_3) δ 161.40 (t, $^2J_{\text{C-F}} = 28.4$ Hz), 135.99, 129.44 (t, $^3J_{\text{C-F}} = 5.5$ Hz), 129.36, 125.88, 124.14, 120.37, 116.72 (t, $^1J_{\text{C-F}} = 256.6$ Hz), 55.84, 45.91, 37.47 (t, $^3J_{\text{C-F}} = 24.4$ Hz), 6.42. $^{19}\text{F NMR}$ (376 MHz, CDCl_3) δ -104.77 (s, 2F). **HRMS** (ESI) calcd for $\text{C}_{14}\text{H}_{18}\text{F}_2\text{NO}_3\text{S}$ $[\text{M} + \text{H}]^+$: 318.0970, found: 318.0974.

$^1\text{H NMR}$ Spectra of **26**

¹³C NMR Spectra of 26

hxy-4-73-11.3.fid

161.68
161.40
161.12
135.99
129.49
129.44
129.38
129.36
125.88
124.14
120.37
119.26
116.72
114.18
77.48
77.16
76.84
55.84
45.91
37.72
37.47
37.23
6.42

¹⁹F NMR Spectra of 26

hxy-4-73-11.2.fid

-104.77

Reply to Reviewer 2 comments:

In this manuscript, Huang and co-workers describe an iron-catalyzed difunctionalization of alkenes with two different alkyl halides via a radical-anion relay strategy. By using Na₂S₂O₄ as both reductant and sulfone-source, this reaction provide a facile method for preparing various dialkyl sulfones with high regio- and chemoselectivity. The authors made much efforts to expand the substrate scope with respect to alkenes and alkyl halides. Control experiments are well performed to illustrate the reaction mechanism.

However, the main content of this protocol is based on the widely explored SO₂ insertion strategy. Moreover, Na₂S₂O₄ serving as the dual role of reductant and sulfone-source has been demonstrated by other groups as well (From Scifinder, a review is appeared from Wu's group: Chin. Chem. Lett. 2021, 32, 461-464.). The main highlight of this manuscript is switching to alkene difunctionalization for the generation of new carbon-centered radicals to be captured by SO₂. It seems that this work is an extension of the previous reports in this field, no matter from the reaction design or the reaction mechanism. In my opinion, this work is not novelty and attractive enough to be published on Nat. Commun. Publication in in a more specialized journal is suggested.

Reply: Thanks for the critical comments. We would respectfully disagree with the comments that this work is not novelty enough. In fact, *this work has provided a new resolution to address the long-standing challenge existed in reductive alkene-difunctionalization reactions.* As we have mentioned in the main text “Due to the presence of alkylmetal species as well as the involvement of classical oxidative addition and reductive elimination in the catalytic cycle, the metal-catalyzed reductive olefin-difunctionalization reaction still mechanistically suffers from undesired β-H elimination and is reluctant to reductive elimination. The remaining challenges include: (1) excess amounts of metal (Zn, Mn, Mg, etc.) were required as reductant to maintain the catalytic cycle, endowing the reaction atom-uneconomical and environment-unfriendly; (2) directing groups were typically required for olefins to inhibit the competing β-H elimination of alkylmetal species with attenuated reductive elimination activity, significantly limiting the substrate scope and reducing the versatility and step-economy of the protocols”. *By using Na₂S₂O₄ as both reductant and sulfone-source, the present work has successfully incorporated two alkyl halides into the alkene to realize the desired alkene difunctionalization.* Secondly, the elusive difunctionalization of alkenes with two different alkyl halides and forging C-C and C-S bonds has been developed by taking advantage of the radical involved electron-shuttle catalysis mechanism, which *illustrated that tuning the catalytic paradigm in metal-catalyzed reductive alkene-difunctionalization reactions may allow to divert mechanistic pathways toward the formation of high value-added products.* In short, the present work not only provides a facile method for preparing various dialkyl sulfones with high regio- and chemoselectivity, but also provides some new insight for developing efficient catalytic paradigm. We believe that these novelty characters are extremely important for chemistry community and meet the publication criteria of this journal.

Reply to Reviewer 3 comments:

We thank **Reviewer 3** for his/her positive comments to improve our manuscript, to which we have responded as follows.

In this manuscript, Huang and co-workers report an unprecedented iron-catalyzed four-component difunctionalization of alkenes to install alkylsulfonyl and fluoroalkyl groups across the double bond in a regioselective fashion. This transformation delivers, in moderate to high yields, a wide array of densely functionalized compounds. This method is operationally simple, exhibits a remarkable and large functional group tolerance and offer a new strategy to access fluorine-containing frameworks of high interest. In addition, the authors demonstrate the important synthetic utility of these compounds through a variety of derivatizations, present a solid mechanistic study that globally support their mechanistic proposal along a well-described the supporting information. Overall, I think that it is an excellent paper that is suitable for publication in NatureComm. However, several points need to be addressed first:

(1) The title should be modified as it is misleading, the authors used the term “alkylative” while “fluoroalkylative” would be more adapted (a single example without fluorine groups installed is presented).

Reply: Thanks a lot for the reviewer’s meticulous inspection. The title has been revised as “Iron-catalyzed fluoroalkylative alkylsulfonylation of alkenes via radical-anion relay”. The accompanying text has been revised as well.

(2) Key references are missing, notably Chem. Eur. J.2019, 25, 1824 that displays relatively similar frameworks while less complex than in the present paper. It should be clearly mentioned in the introduction as prior art The following references should be added too: Green Chem 2023, 25, 9292; Org. Lett. 2020, 22, 2801; Chem 2021, 7, 3412.

Reply: Thanks for your kind suggestions, we are sorry for our carelessness. These references have been added as cited references 32-35 . Moreover, to clearly demonstrate the state of art of the present work, we have added these references and one note as:

Ref 32: Although the fluoroalkylative alkylsulfonylation of olefins has been realized by Liu and coworkers with DABSO as sulfone source, two equivalent zinc have to be utilized as reductant. Liu, Y., Lin, Q., Xiao, Z., Zheng, C., Guo, Y., Chen, Q.-Y. & Liu, C. Zinc-mediated intermolecular reductive radical fluoroalkylsulfonation of unsaturated carbon–carbon bonds with fluoroalkyl bromides and sulfur dioxide. *Chem. Eur. J.* **25**, 1824–1828 (2019).

Other references has been cited as:

Ref 33: Li, J., Guo, Z., Zhang, X., Meng, X., Dai, Z., Gao, M., Guo, S. & Tang, P. Light-induced arylidifluoromethyl-sulfonylation/thioetherification of alkenes using arenethiolates as a photoreductant and sulfur source. *Green Chem.* **25**, 9292–9300 (2023).

Ref 34: Tanaka, S., Nakayama, Y., Konishi, Y., Koike, T. & Akita, M. Fluoroalkanesulfonate salts as dual fluoroalkyl and SO₂ sources: atom-economical fluoroalkyl-sulfonylation of alkenes and alkynes by photoredox catalysis. *Org. Lett.* **22**, 2801–2805 (2020).

Ref 35: Wang, H., Bellotti, P., Zhang, X., Paulisch, T. O. & Glorius, F. A base-controlled switch of SO₂ reincorporation in photocatalyzed radical difunctionalization of alkenes. *Chem* **7**, 3412–3424 (2021).

(3) In Scheme 1, the authors present several bioactive molecules incorporating a sulfone motif. Yet, not a single of them closely resemble the compounds prepared in the present paper. Thus, either the authors remove them since they are not relevant to this study or the authors present molecules that can at least vaguely be linked to their products.

Reply: Thanks for the reviewer’s advice. We have revised the Fig. 1c and some molecules which are more related to this work have been added. Moreover, the intermediates for the synthesis Erysolin and its fluorinated counterparts have been prepared by using our catalytic protocol as a key reaction step.

Fig. 1c

Intermediates for synthesis Erysolin and its fluorinated counterparts

(4) In terms of scope, it seems that styrenes are a limitation. However, this point is not commented by the authors as they only show a single example. They should give a clearer picture of this reactivity by reporting styrenes incorporating strong electron-withdrawing groups (CF_3 for instance) and electron-donating groups (OMe for instance), the simple styrene and a vinyl heteroaromatic.

Reply: The reviewer made an excellent point here. Indeed, aryl alkenes were not well adapted to this reaction. Although we have screened a lot of reaction conditions to improve the reactivity of the aryl alkenes, only styrenes containing electron-rich substituents can give the corresponding products in lower yields. In order to clearly demonstrate this point, we have put one sentence in the main text as: “However, aryl alkenes were not well adapted to this reaction, only styrenes containing electron-rich substituents can give the corresponding products in lower yields (18-20).”

4-(Ethylsulfonyl)-2,2-difluoro-4-(4-methoxyphenyl)-N-phenylbutanamide (18)

The title compound was prepared according to the *General procedure I* and purified by column chromatography as a white solid (73.4 mg, 46% yield). 1H NMR (400 MHz, $CDCl_3$) δ 7.75 (s, 1H), 7.47 – 7.27 (m, 6H), 7.17 (t, $J = 7.8$ Hz, 1H), 6.85 (d, $J = 8.8$ Hz, 2H), 4.41 (dd, $J = 11.2, 3.0$ Hz, 1H), 3.71 (s, 3H), 3.31 – 2.99 (m, 2H), 2.84 – 2.62 (m, 2H), 1.27 (t, $J = 7.5$ Hz, 3H). ^{13}C NMR (101 MHz, $CDCl_3$) δ 161.02 (t, $^2J_{C-F} = 27.8$ Hz), 160.59, 135.73, 130.85, 129.20, 125.80, 123.24, 120.30, 116.79 (t, $^1J_{C-F} = 256.9$ Hz), 114.70, 61.57 (t, $^3J_{C-F} = 3.8$ Hz), 55.36, 44.77, 32.33 (t, $^2J_{C-F} = 24.5$ Hz), 6.23. ^{19}F NMR (376 MHz, $CDCl_3$) δ -101.09 (d, $J = 256.1$ Hz, 1F), -104.54 (d, $J = 256.1$ Hz, 1F). HRMS (ESI) calcd for $C_{19}H_{22}F_2NO_4S$ $[M + H]^+$: 398.1232, found: 398.1230.

¹H NMR Spectra of 18

¹³C NMR Spectra of 18

hxy-4-74-2-20231209_12.fid

¹⁹F NMR Spectra of 18

hxy-4-74-2-20231209_11.fid

-100.75
-101.43
-104.20
-104.88

4-(Ethylsulfonyl)-2,2-difluoro-N,4-diphenylbutanamide (20)

The title compound was prepared according to the *General procedure I* and purified by column chromatography as a white solid (42.5 mg, 29% yield). ¹H NMR (400 MHz, CDCl₃) δ 7.81 (s, 1H), 7.51 – 7.44 (m, 2H), 7.43 – 7.29 (m, 7H), 7.21 – 7.15 (m, 1H), 4.48 (dd, *J* = 11.0, 2.9 Hz, 1H), 3.37 – 3.20 (m, 1H), 3.18 – 3.01 (m, 1H), 2.81 – 2.60 (m, 2H), 1.27 (t, *J* = 7.5 Hz, 3H). ¹³C NMR (101 MHz, CDCl₃) δ 160.85 (t, ²*J*_{C-F} = 27.9 Hz), 135.66, 131.98, 129.71, 129.58, 129.31, 129.26, 125.90, 120.40, 116.72 (t, ¹*J*_{C-F} = 256.5 Hz), 62.17, 44.98, 32.46 (t, ²*J*_{C-F} = 24.3 Hz), 6.22. ¹⁹F NMR (376 MHz, CDCl₃) δ -102.08 (d, *J* = 256.7 Hz, 1F), -103.49 (d, *J* = 256.7 Hz, 1F). HRMS (ESI) calcd for C₁₈H₁₉F₂NNaO₃S [M + Na]⁺: 390.0946, found: 390.0950.

¹H NMR Spectra of 20

¹³C NMR Spectra of 20

hxy-4-74-1-2.13.fid

¹⁹F NMR Spectra of 20

hxy-4-74-1-2, 12, f1.d

-101.74
-102.42
-103.15
-103.83

(5) A question of curiosity: could aryl bromides instead of alkyl ones could be used to generate aryl sulfones?

Reply: In general, the electrophilicity of aryl halides is much lower than that of alkyl counterparts. As a result, the simple aryl bromide is not applicable in the present reaction. However, the modified aryl halides functionalized with an electron-deficient substituent could be utilized as electrophiles to the present reaction: when 1-chloro-4-nitrobenzene was used instead of alkyl ones, the aryl sulfone could be obtained in 43% isolated yield.

2,2-Difluoro-4-methyl-4-((4-nitrophenyl)sulfonyl)-N-phenylpentanamide (**84**)

The title compound was prepared according to the *General procedure I* and purified by column chromatography as a yellow oil (71.3 mg, 43% yield). ¹H NMR (400 MHz, CDCl₃) δ 8.07 (s, 1H), 7.54 (d, *J* = 7.9 Hz, 2H), 7.38 (t, *J* = 7.9 Hz, 3H), 7.22 (t, *J* = 7.4 Hz, 3H), 7.13 (s, 1H), 2.79 (t, *J* = 17.4 Hz, 2H), 1.50 (s, 6H). ¹³C NMR (101 MHz, CDCl₃) δ 161.99 (t, ²*J*_{C-F} = 27.6 Hz), 136.83, 135.58, 130.64, 129.66, 129.44, 126.28, 122.34, 121.01, 117.65 (t, ¹*J*_{C-F} = 257.8 Hz), 63.64, 38.98 (t, ²*J*_{C-F} = 23.3 Hz),

23.33. ^{19}F NMR (376 MHz, CDCl_3) δ -99.49 (s, 2F). HRMS (ESI) calcd for $\text{C}_{18}\text{H}_{18}\text{F}_2\text{N}_2\text{NaO}_5\text{S}$ [$\text{M} + \text{Na}$] $^+$: 435.0797, found: 435.0804.

^1H NMR Spectra of **84**

^{13}C NMR Spectra of **84**

¹⁹F NMR Spectra of **84**

hxy-4-75-3.11.fid

-99.49

(6) Regarding the mechanism, the authors concluded that the last step occurred via a nucleophilic substitution because the bromomethylcyclopropane does not open. However, this compound is not fully activated, does they get the same result with a phenyl-substituted one?

Reply: To follow the reviewer's suggestion, we have synthesized the compound **S3-20** according to the known method (*J. Med. Chem.* **2016**, 59, 3215–3230). However, when the **S3-20** was used as a coupling partner, neither the desired product **A** nor the ring-opening product **A'** was obtained. Instead, by-products **B**, **C** (*Nat. Chem.* **2019**, 11, 1158–1166), and **D** (*Org. Lett.* **2010**, 12, 5370–5373) were observed. According to previous reports (*Tetrahedron* **2019**, 75, 4486–4496; *J. Am. Chem. Soc.* **2019**, 141, 6853–6858; *Org. Lett.* **2010**, 12, 5370–5373), the reason might be attributed to the higher reactivity of **S3-20** as it was easily to accept an electron from Na₂S₂O₄ to generate the radical intermediate **I** and **II**. These experimental results further supported that the final step of the present protocol took place via nucleophilic substitution.

1. Synthesis of substrate **S3-20**

2. Standard conditions

2,2-Difluoro-4-methyl-N-phenylpentanamide (B)

¹H NMR (400 MHz, CDCl₃) δ 8.14 (s, 1H), 7.55 (dd, *J* = 8.6, 1.2 Hz, 2H), 7.36 (t, *J* = 7.9 Hz, 2H), 7.22 – 7.15 (m, 1H), 2.46 (t, *J* = 17.2 Hz, 2H), 2.40 – 2.19 (m, 1H), 1.36 (s, 6H). ¹³C NMR (101 MHz, CDCl₃) δ 162.91 (t, ²*J*_{C-F} = 28.2 Hz), 136.05, 129.31, 125.81, 120.59, 117.95 (t, ¹*J*_{C-F} = 254.2 Hz), 68.92 (t, ³*J*_{C-F} = 3.8 Hz), 45.77 (t, ²*J*_{C-F} = 21.3 Hz), 30.59 (t, ⁴*J*_{C-F} = 1.9 Hz). ¹⁹F NMR (376 MHz, CDCl₃) δ -100.27 (s, 2F). HRMS (ESI) calcd for C₁₂H₁₆F₂NO [M + H]⁺: 228.1194, found: 228.1185.

(1-Bromobut-3-en-1-yl)benzene (C)

¹H NMR (400 MHz, CDCl₃) δ 7.44 – 7.38 (m, 2H), 7.38 – 7.32 (m, 2H), 7.32 – 7.28 (m, 1H), 5.87 – 5.67 (m, 1H), 5.22 – 5.06 (m, 2H), 4.96 (t, *J* = 7.5 Hz, 1H), 3.16 – 2.79 (m, 2H). ¹³C NMR (101 MHz, CDCl₃) δ 134.87, 128.83, 128.57, 127.50, 125.95, 118.29, 54.23, 44.18.

(E)-Buta-1,3-dien-1-ylbenzene (D)

¹H NMR (400 MHz, CDCl₃) δ 7.40 (d, *J* = 7.6 Hz, 2H), 7.31 (t, *J* = 7.5 Hz, 2H), 7.23 (d, *J* = 6.9 Hz, 1H), 6.79 (dd, *J* = 15.6, 10.5 Hz, 1H), 6.62 – 6.40 (m, 2H), 5.33 (d, *J* = 16.9 Hz, 1H), 5.17 (d, *J* = 9.9 Hz, 1H). ¹³C NMR (101 MHz, CDCl₃) δ 137.31, 137.24, 132.98, 129.74, 128.74, 127.76, 126.57, 117.77.

¹H NMR Spectra of B

hxy-qinghua-20230424.10.fid

¹³C NMR Spectra of B

hxy-qinghua-20230424.12.fid

¹⁹F NMR Spectra of B

hxy-qinghua-20230424_11.fid

¹H NMR Spectra of C

¹³C NMR Spectra of C

hxy-4-sanyuanhuan-Br, 11, fid

¹H NMR Spectra of D

^{13}C NMR Spectra of **D**

hxy-4-81-3.11.fid

(7) I do not know if it is due to my computer, but NMR spectra appear a bit blurry and are not always easy to look at. Please fix this issue.

Reply: We are sorry for our carelessness. The blurriness of the NMR spectra is caused by over-compression of the file. We have revised the file and the clear NMR spectra have been put into the revised SI.

REVIEWERS' COMMENTS

Reviewer #1 (Remarks to the Author):

The authors have fully addressed my concerns in the previous review. I recommend acceptance of the paper.

Reviewer #2 (Remarks to the Author):

I insist on my previous reports, since this work lacks of novelty and significance. I cannot support its publication in Nature Commun.

Reviewer #3 (Remarks to the Author):

In this revised version, the authors answered to all my comments and the ones of the other reviewers. They performed all the reactions suggested, adding several new results, and made all the corrections required. Thus, I am fully satisfied with this new version and recommend publication as it is.